# Efficient Separation of Heavy Metals by Magnetic Nanostructured Beads

**Lisandra de Castro Alves** , **Susana Yáñez-Vilar \*** , **Yolanda Piñeiro-Redondo** **and José Rivas**

Applied Physic Department, NANOMAG Laboratory, Research Technological Institute, Universidade de Santiago de Compostela (USC), 15782 Santiago de Compostela, Spain; lisandracristina.decastro@usc.es (L.d.C.A.); yolanda.fayoly@gmail.com (Y.P.-R.); jose.rivas@usc.es (J.R.)

\* Correspondence: susana.yanez@usc.es

**Abstract:** This study reports the ability of magnetic alginate activated carbon (MAAC) beads to remove Cd(II), Hg(II), and Ni(II) from water in a mono-metal and ternary system. The adsorption capacity of the MAAC beads was highest in the mono-metal system. The removal efficiency of such metal ions falls in the range of 20–80% and it followed the order Cd(II) > Ni(II) > Hg(II). The model that best fitted in the ternary system was the Freundlich isotherm, while in the mono-system it was the Langmuir isotherm. The maximum Cd(II), Hg(II), and Ni(II) adsorption capacities calculated from the Freundlich isotherm in the mono-metal system were 7.09, 5.08, and 4.82 $(mg/g) (mg/L)^{1/n}$, respectively. Lower adsorption capacity was observed in the ternary system due to the competition of metal ions for available adsorption sites. Desorption and reusability experiments demonstrated the MAAC beads could be used for at least five consecutive adsorption/desorption cycles. These findings suggest the practical use of the MAAC beads as efficient adsorbent for the removal of heavy metals from wastewater.

**Keywords:** heavy metals; magnetite nanoparticles; adsorption; nanocomposite; hybrid; multi-metal; water

## 1. Introduction

Water pollution by heavy metals has become a serious problem due to the adverse effects on ecosystems and human health. More specifically, cadmium (Cd), mercury (Hg), lead (Pb), or nickel (Ni) are known to be highly carcinogenic and mutagenic at low concentrations, and may produce acute toxicity or even dead in living organisms, when present slightly above their allowed limits [1–3]. Although diverse technologies have been developed for the removal of heavy metals from water sources [4], there is still an urgent need for facile cleaning procedures that ensure high efficiency in the low concentration ranges. Chemical precipitation, ion precipitation, ion exchange, and adsorption are some of the most used techniques due to their potential for scaling up. Among them, the use of natural biopolymers, such as alginate, agarose, chitin and pectin [5–7] in metal biosorption from wastewaters has gained much attention in recent years. Alginate is a polysaccharide derived from brown algae and in the majority of the studies it has been used in the form of calcium alginate beads, due to its practical handling [8]. Nevertheless, the separation of the loaded biomaterial from the medium is often a problem. To overcome this problem, magnetite nanoparticles ($Fe_3O_4$-NPs) are being incorporated onto the biosorbent matrix [9–11] giving the possibility to magnetically manipulate and separate the hybrid materials from the water matrix. Magnetic alginate beads are a very attractive material with multiple properties such as high specific surface area, rapid recovery, cost-effectiveness, and chemical versatility, for which they are amenable to be combined with materials to increase their affinity for pollutants. Humic acid [12], Cyanex 302 [12] and microalgae [7] were incorporated into alginate beads for their

affinity to metal ions. Different authors have shown the ability of several alginate beads composition to uptake metal ions from aqueous solutions. It was observed that magnetic alginate beads containing silica coated with iron carbide nanoparticles enhanced more the adsorption of copper ions than alginate beads alone [13]. In addition, it was studied that magnetic nanoparticles functionalized with citrate ions present an enhanced adsorption of Pb(II) metal ions from solution [14]. In one of our previous works [15], magnetic alginate beads tailored with commercial activated carbon revealed a high capacity for cadmium ions uptake. A 35% removal percentage of cadmium ions was achieved over 1 h with less than 15 mg of adsorbent used. These nanostructured beads revealed to be a great assessment for industrial use, for their high adsorption surface area, easy handling, and magnetic separation from any aqueous media. For these reasons, the same nanostructured beads were used in the present study and tested under more realistic conditions by studying their adsorption capacity when exposed to a mixture of heavy metals. With this aim, different adsorption tests were performed on mono-metal and ternary systems, comprising Cd(II), Hg(II) and Ni(II) metals ions, which are commonly found in waste water from industry and mining effluents. To gain insights in the adsorption mechanism other relevant aspects (e.g., metal-adsorbent mechanisms, metal distribution on beads surface and internal structure, metal desorption, reuse, porosity, among others) were studied in detail.

## 2. Results and Discussion

### 2.1. Morphology of Magnetic Beads

The morphology of the MAAC beads and commercial activated carbon was examined using scanning electron microscopy (SEM) and transmission electron microscopy (TEM). In Figure 1a, SEM micrograph shows that commercial activated carbon presents a disordered layer-like structure, which was further studied with TEM, reveals a wide variation in the layer size range between 40 µm to 320 µm (Figure 1b). SEM micrograph (Figure 1c) of a representative MAAC bead shows a spherical morphology with a porous and layer-like structure on the surface inherited from the precursor commercial activated carbon. The internal structure, as can be observed in Figure 1d, is a combination of activated carbon layers within the interconnected porous network of alginate in MAAC beads.

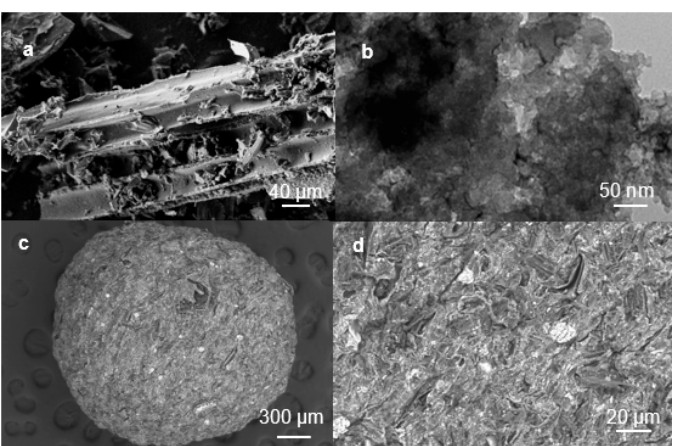

**Figure 1.** Commercial activated carbon SEM (**a**) and TEM (**b**) micrograph. SEM image of the MAAC bead surface (**c**) and internal structure (**d**).

### 2.2. Structural and Textural Characterization

The X-ray diffraction pattern of the MAAC beads in Figure 2a shows the presence of sharp diffraction peaks located at $2\theta$ = 30.1, 35.5, 43.2, 53.5, 57.1, 62.7°. This is the characteristic diffraction pattern arising from the reflection of planes (022), (113), (004), (224), (115) and (044) corresponding to crystalline $Fe_3O_4$-NPs embedded in the porous beads. The MAAC bead's surface was also analyzed by Fourier transform infrared (FT-IR) spectroscopy. As shown in Figure 2b, the peaks at 3228 cm$^{-1}$ and

1076 cm$^{-1}$ are related to the –OH and –C–O stretching vibration bands of alginate, respectively, while peaks at 1585 and 1286 cm$^{-1}$ are attributed to the asymmetric and symmetric stretching vibrations of the carboxyl groups of alginate, respectively [16]. Finally, the band at 558 cm$^{-1}$ is due to collective vibrations of the magnetite lattice [17], which confirms the successful incorporation of Fe$_3$O$_4$-NPs into the alginate matrix.

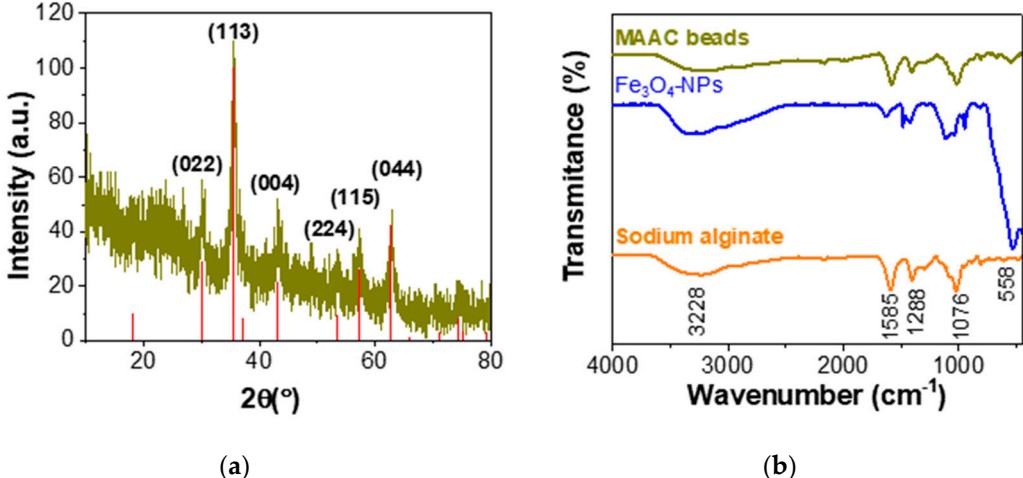

(**a**)                                          (**b**)

**Figure 2.** (**a**) X-ray diffraction and infrared spectra and (**b**) IR spectra of MAAC beads.

The mean particle size of the Fe$_3$O$_4$ nanoparticles employed for the synthesis of the MAAC beads were calculated from the X-ray diffraction (XRD) pattern (Figure S1) according to the linewidth of the (113) plane refraction peak using Scherrer equation:

$$L = K\lambda/\beta\cos\theta \tag{1}$$

where L is the mean size of the ordered (crystalline) domains, λ is the X-ray wavelength, β is the width of the XRD peak at half height, K is a shape factor, about 0.9 for magnetite and θ is the Bragg angle The particle diameter calculated from the X-ray diffractogram, using the Scherrer equation was 9.58 nm. By TEM the image (Figure S1) we can observe nanoparticles with a size of 20 nm. The difference between TEM and XRD can be attributed to the fact that "crystallite size" is not synonymous with "particle size", while XRD is sensitive to the crystallite size inside the particles.

### 2.3. Specific Surface Are and Pore-Distribution

The textural properties of commercial activated carbon (commercial AC) and MAAC beads were analyzed with Brunauer–Emmett–Teller (BET) porosimetry and shown in Table 1. The surface area of commercial AC (849.32 m$^2$/g) is within the theoric range for activated carbons (500 to 3.000 m$^2$/g) [8]. The surface area of the MAAC beads is 107.13 m$^2$/g which is two orders of magnitude higher than the reported 6.25 m$^2$/g for calcium alginate beads [18], and can be ascribed to the presence of activated carbon. The pore volume of the MAAC beads (0.075 cm$^3$/g) was smaller than the one of commercial AC (0.28 cm$^3$/g). In addition, the pore diameter size of the MAAC beads (1.24 nm) is slightly smaller than the commercial AC (1.26 nm). This could be attributed to the deposition of the commercial AC on the surface of the MAAC beads (both surface and internally), leading to a complete filling of the smaller pores.

**Table 1.** Textural parameters of Commercial AC and MAAC beads.

| Samples | BET Surface Area ($S_{BET}$) [1] (m$^2$/g) | Pore Volume (cm$^3$/g) | Pore Diameter [2] (nm) |
|---|---|---|---|
| Commercial AC | 849.32 | 0.28 | 1.26 |
| MAAC | 107.13 | 0.075 | 1.24 |

[1] $S_{BET}$ is the BET surface area evaluated at a relative pressure (p/p0) of 0.99. [2] Pore diameter calculated using the Barrett–Joyner–Halenda (BJH) method.

## 2.4. Magnetic Properties of MAAC Beads

Figure 3a shows the variation of magnetization, M, as a function of temperature of MAAC beads in the range 5 to 350 K in an external magnetic field of 100 Oe recorded in zero-field cooling (ZFC) and field cooling (FC). From the curves it is clearly observed the superimposition of the ZFC and FC curves take place at 275 K. The superimposition of ZFC and FC curves is one of the characteristic features of a superparamagnetic system. The magnetic content on the MAAC beads was calculated by thermogravimetric analyses (TGA), which was equal to 23%. Figure 3b illustrates the magnetization curves of bare $Fe_3O_4$-NPs and of the MAAC nanocomposite beads. The saturation of magnetization (Ms) for the synthesized $Fe_3O_4$-NPs (69.23 emu/g) was higher than the observed for the MAAC beads (48.62 emu/g). This may be attributed to the coating effect of alginate trapping the $Fe_3O_4$-NPs in the gel matrix. However, the MAAC beads have superparamagnetic behavior and are easily separated from solution with the help of an external magnetic force.

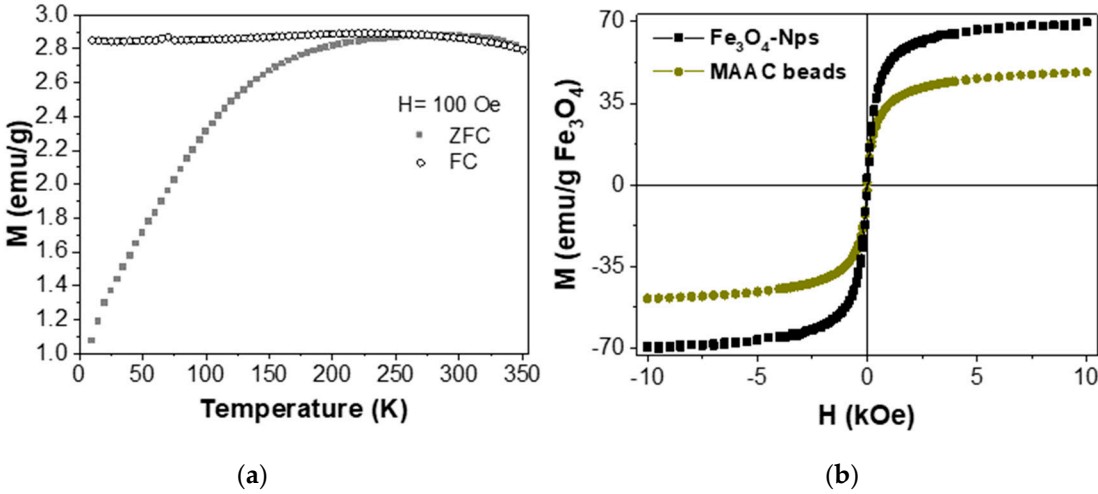

(**a**)  (**b**)

**Figure 3.** ZFC and FC curve recorded at 100 Oe (**a**) and magnetization curve of MAAC bead and $Fe_3O_4$-NPs at 25 °C (**b**).

## 3. Adsorption Study

### 3.1. Desorption and Reusability

The reusability of the MAAC bead was study by repetitive adsorption and desorption cycles of Cd(II) metal ions using 0.01 M HCl solution, as the desorption solution. The q (mg/g) desorption of cadmium was calculated directly from the amount of cadmium adsorbed and amount of cadmium desorbed using Equation (2):

$$q(desorption) = \frac{(M_{adsorb} - M_{desorb})V}{M} \tag{2}$$

where $M_{adsorb}$ and $M_{desorb}$ are the adsorbed and desorbed amount of metal ions (mg/g), respectively; $V$ (L) is the volume of desorption solution and $M$ (g) the mass of the MAAC beads used. The desorption percentage was calculated using the following Equation (3):

$$\% \text{ Desorption} = \frac{M_{desorb}}{M_{adsorb}} \times 100 \tag{3}$$

The desorption of cadmium from MAAC bead showed a fair desorption percentage over the five cycles, as can be observed in Figure 4. However, the adsorption capacity decreased with increasing regeneration cycle number, except for on cycle number two. The cadmium metal ions have entered inside of the beads structure after the first cycle (Figure S2) resulting in the maximum adsorption capacity on cycle two. Continuous adsorption resulted on internal pore saturation and consequently, on the decreased of adsorption capacity and increase amount of desorbed cadmium metal ions found in solution. Furthermore, the results suggest a reduction of the metal ions from the matrix in the first fourth cycles and more than 50% of metal was recovered. After the fifth cycle, the desorption percentage decreased to 20%, since cadmium metal ions remained inside of the beads structure after the third desorption cycle. Thus, it can be said that these beads have the potential to be reuse up to four cycles under the chosen conditions. For further studies, factors, such as time and desorbent concentration must be considered for maximum desorption capacity for more than 5 cycles. In previously studies [19], hydrochloric acid proved to be a good desorbent within 2 h of repetitive adsorption and desorption cycles.

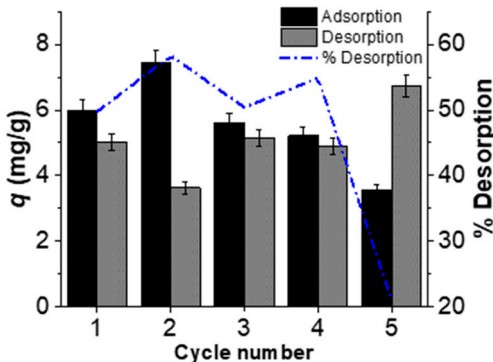

**Figure 4.** Adsorption and Desorption cycles of cadmium from MAAC beads (mean ± SE of 15 replicates).

### 3.2. Effect of pH

The influence of pH on Cd(II), Hg(II) and Ni(II) adsorption capacity by the MAAC beads was studied at pH range of (2.0 to 9.0). In Figure 5, the adsorbed metal ions per adsorbent mass q (mg/g) are presented for all tested pH values. As can be seen, the adsorption process for Cd(II) and Ni(II), was constant between the pH values of 4.5 to 7.0, followed by an increase at higher pH values. On contrary, Hg(II) adsorption depends in a non-predictable way on the pH, attaining a maximum adsorption capacity at pH 4.5, followed by a steep decrease at pH 6.5 and a large increment up to pH 9.0. The higher concentration of hydrolyzed ions such as $H^+$ allow an enhanced binding of Hg(II) metal ions to the sodium alginate surface [19]. Besides this binding mechanism being more enhanced at pH 2.0 for Hg(II) ions, at pH 4.5 all metal ions revealed to have a constant and high adsorption capacity at this pH, being the selected for the experiments at the mono and ternary system. The increasing adsorption at higher pH values for all metal ions may be attributed to the formation of hydroxyl ions [20].

At higher pH values, the decreased adsorption capacity by Cd(II) coincides with the decreasing concentration of Cd(II) and the precipitation of $Cd(OH)_2$ into solution. The $Cd(OH)_2$ ionic species are adsorbed by the MAAC beads occupying the available sites and preventing the further adsorption of Cd(II) ions [21]. For nickel an increasing trend at higher pH values was observed. This characteristic trend is attributed to the specific adsorption of cationic hydroxo-complexes, which is the pH range

more favorable for the formation of these species [22]. The hydrolysis of mercury, on the other hand, begins at very low pH values (pH < 4.5) with the formation of $Hg(OH)^+$ and at pH values (pH > 4.5) the adsorption capacity suddenly increases because of the formation of mercury neutral species $Hg(OH)_2$ [23].

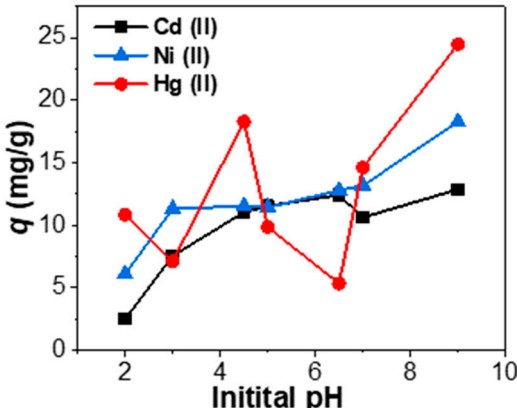

**Figure 5.** Effect of initial pH on Cd(II), Hg(II) and Ni(II) adsorption onto MAAC beads (mean standard deviation ± 0.3).

### 3.3. Adsorption of Mono and Ternary Systems

Batch adsorption experiments were performed using a defined amount of MAAC beads in mono and ternary systems containing Cd(II), Ni(II) and Hg(II) metal ions. The effect of the initial metal ion concentration on adsorption was studied for solutions prepared with a set of concentrations 10 to 250 mg/L, and applying previously optimized conditions (magnetic agitation at 300 rpm, using 14 mg of adsorbent during 6 h) at fixed pH = 4.5. The equilibrium adsorption capacity ($q_e$) was calculated according to the following equation [24]:

$$q_e = \frac{(C_0 - C_e)V}{M} \tag{4}$$

The removal efficiency ($R\%$) of Cd(II), Hg(II) and Ni(II) in the mono-component system was calculated using the following equation:

$$R\% = \frac{(C_0 - C_e)}{C_0} \times 100 \tag{5}$$

where $q_e$ is the equilibrium adsorption capacity (mg/g); $C_0$ and $C_e$ are the initial and equilibrium concentration (mg/L) of metal ions, respectively; $V$ is the volume of working solution (L) and $M$ is the weight (g) of adsorbent used. Alternatively, $q$, can be expressed in terms of molarity, $q$ (mol/g), to gain insights into the number of moles (atoms) that adsorb on the cleaning beads (Figure S3).

In Figure 6, the adsorption capacity, $q$ (mg/g), of Cd(II), Ni(II) and Hg(II) by the MAAC beads, for the mono and ternary metal adsorption batch tests are presented versus the initial concentration in the batch solution of each metal. From all tested metals under the current experimental conditions, it is evident that Cd(II) ions are preferentially adsorbed in both tests, being in the mono-metal case (Figure 6a) the adsorption capacity of Cd(II) ions twice the adsorption observed for Ni(II) and Hg(II).

In addition, in the mono-metal test, the adsorption of cadmium and nickel ions increases uniformly indicating ongoing adsorption process onto the MAAC beads, on contrary to mercury, which shows a decreased adsorption at the initial concentration of 150 mg/L. In the ternary metal system (Figure 6b), the adsorption capacity is generally smaller than in the mono-metal case, indicating a competition between the metal ions in the ternary system for available binding sites on the MAAC beads [25], with a striking exception at initial metal concentration of $C = 150$ mg/L, where all metal ions are adsorbed

with more efficacy. Moreover, the similarities of the adsorption capacity curve of all ions in the ternary system with the mercury adsorption feature in the mono-metal system suggests that mercury has an important role modulating the adsorption mechanism, which will deserve future studies.

The removal efficiency decreases when the initial metal ions concentration is increased (Figure 7), a trend that was already observed in a previous study [15]. This behavior is mainly ascribed to the saturation of the available binding sites during the adsorption process, leading to a reduction of the adsorption capacity.

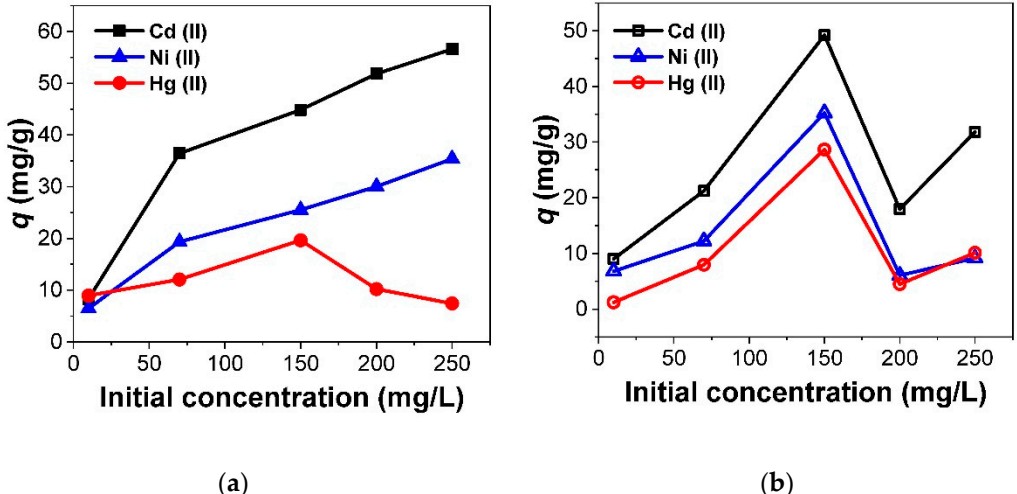

(**a**)                                            (**b**)

**Figure 6.** Effect of initial metal concentration on the adsorption capacity of MAAC beads for (**a**) mono-metal and (mean SD ± 1.45) (**b**) ternary experiment at pH 4.5 (mean SD ± 1.27).

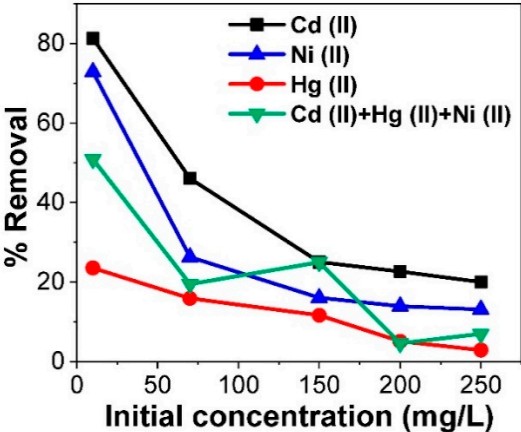

**Figure 7.** Removal efficiency or mono-metal and ternary experiments at pH 4.5.

### 3.4. Competitive Adsorption Evaluation

The interactive effect of Cd(II), Hg(II) and Ni(II) was investigated in the ternary system. For that, an evaluation ratio was introduced to assess the type of adsorption competition between each metal ion in the system and combined [26]. The evaluation ratio is expressed by the following Equation (6):

$$E = \frac{Q'_e}{Q_e} \qquad (6)$$

where $E$, is the evaluation ratio; $Q'_e$ (mg/g) is the amount of metal ions adsorbed in a ternary system and $Q_e$ (mg/g) is the amount of metal ions adsorbed in the mono-metal system. If the evaluation ratio, $E > 1$, the presence of other metal ions have enhanced the adsorption of other metal ions in solution (synergism effect); when $E = 1$, this means that the presence of other metal ions would not influence the

adsorption of another metal ion; and when $E < 1$, the presence of another metal ion would suppress the adsorption of one another (antagonism effect). The evaluation ratios of individual Cd(II), Hg(II) and Ni(II) in the ternary system and the whole mixture are listed in Table 2.

**Table 2.** Individual and sum of the evaluation ratios of Cd(II), Hg(II) and Ni(II) in the ternary system for MAAC beads.

| Initial Concentration (mg/L) | | 10 | 70 | 150 | 200 | 250 |
|---|---|---|---|---|---|---|
| Adsorbent | Metal Ions | Evaluation Ratios | | | | |
| MAAC | Cd(II) | 1.08 | 0.58 | 1.10 | 0.35 | 0.56 |
| | Hg(II) | 0.14 | 0.66 | 1.46 | 0.44 | 1.36 |
| | Ni(II) | 1.05 | 0.63 | 1.38 | 0.20 | 0.26 |
| | Cd(II) + Hg(II) + Ni(II) | 0.68 | 0.71 | 1.48 | 0.32 | 0.48 |

Cadmium and nickel in both mono and ternary systems were most efficiently adsorbed than mercury (Figure 6a,b). On contrary, the study of the evaluation ratios in terms of competitive scenario reveals that mercury ions are the least suppressed of all three metal ions. The results compiled in Table 2, reveal an antagonistic effect ($E < 1$) almost in all concentrations in the individual metals and on the whole mixture of the ternary system, except at $C = 150$ (mg/L), where $E = 1.48$ ($E > 1$), indicating an average synergistic behavior between the metal ions. For high values of initial metal concentrations of $C = 200$ (mg/L) and $C = 250$ (mg/L), the degree of suppression is large for nickel and cadmium ions, while mercury adsorption is favored by a synergistic effect. Moreover, for $C \geq 70$ (mg/L) mercury is the least suppressed metal and benefits from an enhancement effect, in part due to the increase electrostatic repulsion among the cations that would limit the adsorption of the metal ion [27].

### 3.5. Adsorption Isotherms

To determine the adsorption capacity of the MAAC beads in a mono-metal and ternary system, adsorption studies with initial concentrations ranging from $C = 10$ (mg/L), to $C = 250$ (mg/L), were carried out. The adsorption equilibrium was studied fitting the experimental data to the linear equations of Langmuir and Freundlich isotherm models.

Langmuir isotherm model [28]:

$$\frac{C_e}{q_e} = \frac{1}{K_L q_m} + \frac{C_e}{q_m} \tag{7}$$

Freundlich isotherm model [29]:

$$\log q_e = \log K_F + \frac{1}{n} \log C_e \tag{8}$$

where $q_e$ (mg/g) is the amount of metal ions adsorbed; $C_e$ (mg/L) is the adsorbate concentration in solution, both at equilibrium; $K_L$ (L/mg) is the Langmuir adsorption constant; and $q_m$ (mg/g) is the maximum adsorption capacity for monolayer formation on the adsorbent. The value $K_F$ can be defined as the adsorption or distribution coefficient and represents the quantity of metal ions adsorbed onto the beads. The value of $1/n$ indicates surface heterogeneity, which becomes more heterogeneous as its value gets closer to zero. A fundamental characteristic of the Langmuir isotherm is to predict the affinity between sorbate and sorbent using a dimensionless constant, known as separation factor $R_L$, which can be represented as:

$$R_L = \frac{1}{1 + K_L C_0} \tag{9}$$

where $C_0$ (mg/L) is the adsorbate initial concentration. The value of $R_L$ stands between 0 and 1 for favorable adsorption, while $R_L > 1$ represents unfavorable adsorption, $R_L = 1$ represents linear adsorption and $R_L = 0$ for irreversible adsorption processes [28].

Table 3 lists the parameters of Langmuir and Freundlich isotherms models computed from the experimental tests using MAAC beads for mono-metal and ternary systems adsorption.

**Table 3.** Langmuir and Freundlich isotherm parameters.

| Adsorption System | Metal Ions | Langmuir Parameters | | | | Freundlich Parameters | | | |
|---|---|---|---|---|---|---|---|---|---|
| | | $q_m$ (mg/g) | $K_L$ (L/mg) | $R_L$ | $R^2$ | $n$ | $1/n$ | $K_F$ (mg$^{1-(1/n)}$L$^{1/n}$g$^{-1}$) | $R^2$ |
| Cd(II) | Cd(II) | 59.17 | 0.048 | 0.103 | 0.981 | 2.52 | 0.397 | 7.09 | 0.983 |
| Hg(II) | Hg(II) | 25.00 | 0.040 | 0.118 | 0.922 | 3.94 | 0.254 | 5.08 | 0.865 |
| Ni(II) | Ni(II) | 37.04 | 0.031 | 0.144 | 0.945 | 2.79 | 0.358 | 4.82 | 0.995 |
| Cd(II) + Hg(II) + Ni(II) | Cd(II) | 56.18 | 0.029 | 0.339 | 0.758 | 2.49 | 0.401 | 6.11 | 0.889 |
| | Hg(II) | 181.82 | 0.007 | 0.164 | 0.681 | 1.50 | 0.668 | 3.35 | 0.901 |
| | Ni(II) | 172.71 | 0.001 | 0.456 | 0.561 | 1.14 | 0.878 | 0.35 | 0.953 |

The models of Langmuir and Freundlich equations in general described the data well, although the mono-metal system seems to be better described by Langmuir while the ternary system by the Freundlich isotherm, as suggested by the correlation coefficient values $R^2$. The Langmuir constant, $K_L$ values were higher for the mono-metal system, revealing a higher adsorption capacity. Contrary to the ternary system, where the presence of other metal ions in the system decreased the adsorption capacity due to the competition of available adsorption sites onto the MAAC beads. The $R_L$ parameter values stand below 1, indicating favorable and weakly reversible adsorption of studied metal ions onto MAAC beads. The values of $n$ determined with the Freundlich equation were generally higher than 1.0, indicating heterogeneous adsorption process for all metal ions onto the beads. Among all metals, Cd(II) was the most highly adsorbed with a Freundlich $K_F$ constant of 7.09 and 6.11 for both mono-metal and ternary system, respectively. In our study, the adsorption cannot be simply related to the physicochemical properties of metal cations, since cadmium with higher atomic weight and ionic radius than nickel was more intensely adsorbed [30]. Thus, adsorption in the ternary system was better described by the Freundlich adsorption isotherm, revealing heterogeneous surface with different affinity sites on MAAC beads.

## 4. Materials and Methods

### 4.1. Synthesis of MAAC Beads

The procedure of the MAAC beads synthesis is described in our previous study [15]. Briefly, $Fe_3O_4$-NPs were synthesize by reverse coprecipitation method [16]. In the synthesis, 15 mL of 1.0 M $FeCl_3 \cdot 6H_2O$ (Alfa Aesar, Madrid, Spain) and 0.5 M $FeSO_4 \cdot 7H_2O$ (Sigma, St. Louis, MO, USA) were mixed and added dropwise into a 3.5 M $NH_4OH$ solution of 20 mL at 60 °C. The reaction proceeded for 30 min under mechanical agitation. The magnetic nanoparticles were then washed and re-dispersed in distilled water.

Beads were prepared in cross-linking solution using calcium chloride solutions as the cross-linking agent. Next, 2.0 g of sodium alginate (Sigma, St. Louis, MO, USA) was subsequently added to the previously prepared $Fe_3O_4$-NPs solution (35 mL). After obtaining a homogeneous solution, 3.0 g of commercial activated carbon was added and the solution was mechanically agitated for 4 h. The obtained suspension was added dropwise into a previously prepared bath of 0.13 M $CaCl_2$ (Sigma, St. Louis, MO, USA) and 450 μL Tween 20 (Fluka, Steinheim, Germany) under continuous magnetic speed of 450 rpm using a New Era NE-300 syringe pump (Biogen, Madrid, Spain). Beads were instantly formed and were left in the bath around 30 min for hardening. Afterwards, the beads were collected with a magnet and cleaned with distilled water. Finally, the magnetic beads were dried at 60 °C.

### 4.2. Effect of pH

The effect of pH on adsorption capacity of Cd(II), Hg(II) and Ni(II) metal ions was conducted individually by mixing (14 mg) of adsorbent with 20 mL of 10 mg/L$^{-1}$ metal ions concentration. The adsorption capacity was studied at pH values of (2.0, 3.0, 4.5, 5.0, 6.5, 7.0, 9.0) under magnetic

agitation (300 rpm) at ambient temperature for 6 h. The pH values were adjusted with 0.01 M HCl and 0.05 M NaOH using a Milwaukee pH51 waterproof (Aldo, Madrid, Spain). After that, the metal concentration present in the supernatant was determined.

### 4.3. Adsorption Study

Parameters such as adsorbent dosage, rotation speed and agitation type were already optimized in a previous study [16]. The effect of the initial metal concentration was studied with the following conditions (magnetic agitation of 300 rpm, 14 mg adsorbent dosage and contact time of 6 h). A set of solutions with a volume of 15 mL and varying metal concentration between 10 to 250 mg/L were prepared. The pH value was adjusted for each metal with 0.05 M HCL and 0.02 M NaCl using a pH meter Milwaukee pH51 waterproof (Aldo, Madrid, Spain). For the ternary adsorption system, solutions with equal concentrations (mg/L) of Cd(II), Ni(II) and Hg(II) were prepared with the same above experimental conditions used.

### 4.4. Desorption and Reusability

For the desorption experiments, hydrochloric acid as desorbent solution was used. First, the adsorption experiment was carried out by preparing 15 replicates of 20 mL of 10 mg/L solution of Cd(II), using (14 mg) of adsorbent. The solutions were magnetically agitated (300 rpm) for 1 h at ambient temperature. Before performing desorption, the beads were magnetically separated from solution and washed with distilled water. Subsequently, the beads were added to the eluting solution. The desorption was carried out by mixing the beads with 20 mL of 0.01 M HCl solution for 30 min under ultrasonic agitation. In the end of each adsorption and desorption cycles, the concentration of Cd(II) on the supernatant was measured. The beads surface and internal structure was analyzed by energy dispersive x-ray spectroscopy (EDX) mapping on each cycle for adsorption and desorption.

## 5. Characterization

The XRD measurements were performed using a Philips diffractometer (Panalytical, Callo End, UK) with Cu K$\alpha$ radiation ($\lambda$ = 1.5406 Å), with a step size of 0.02° and a counting time of 2 s per step from 10° to 80° (2$\theta$). The Fourier transform infrared spectra (FT-IR) was recorded on a Varian FT-IR 670 (Varian, Palo Alto, CA, USA) spectrophotometer in the range 400–4000 cm$^{-1}$. The morphology of the activated carbon and hybrid magnetic beads were characterized by TEM using a JEOL JEM-1011 microscope (JEOL, Tokyo, Japan) operating at 100 kV and by SEM analysis using a ZEISS FE-SEM ULTRA Plus microscope operated at 30 kV (Zeiss, Oberkochen, Germany). The BET specific surface area and the pore size distribution of the samples were characterized under $N_2$ adsorption-desorption isotherms at 77 K using Micromeritics ASAP 2020 instrument (Micromeritics, Norcross, GA, USA). The TGA were studied using Perkin Elmer Pyris 7 (Perkin, Waltham, MA, USA) under an oxygen flow (20 mL/min) with heating rate of 50 °C up to 840 °C. A superconducting quantum interference device (SQUID) magnetometer (Quantum Design, Darmstadt, Germany) was used to analyze the magnetic properties of the MAAC beads. The FC and ZFC measurements were recorded at an applied field of 100 Oe by scanning between 5 and 350 K. Magnetic properties were assessed by measuring the magnetization curve using a vibrating sample magnetometer (VSM) (DMS, Massachusetts, MA, USA) with an applied field between −10 and 10 kOe at room temperature. The concentration of Cd(II), Hg(II) and Ni(II) ions was measured by inductively coupled plasma optical mission spectrophotometry (ICP-OES) using an emission spectrometer Perkin Elmer Model Optima 3300 DV (Perkin, Waltham, MA, USA).

## 6. Conclusions

Hybrid magnetic beads made by encapsulation of magnetite nanoparticles with sodium alginate and commercial activated carbon proved to be effective on the adsorption of Cd(II), Hg(II) and Ni(II) metal ions. The MAAC beads present a superparamagnetic behavior, although with a moderate saturation magnetization around 50 emu/g, their efficient magnetic separation from solution was allowed.

Their magnetic functionality is a crucial design parameter for industrial applications in which remote manipulation and extraction enhances the decontamination procedure. The quantitative studies on the equilibrium adsorption in the ternary metal system revealed a competitive adsorption process established between the different metal ions. The adsorption affinity, Cd(II) > Ni(II) > Hg(II), was clearly established in both mono and ternary system. The equilibrium for both systems was generally better described by the Freundlich model, indicating an heterogenous adsorption process. Cadmium metal ions were the most adsorbed by the MAAC beads, and consequently, it was the metal studied for the desorption experiment. The desorption rate was significant using hydrochloric acid and the beads tolerate well the desorption process without structure damage. These results support the potential application of the MAAC beads for the removal and recovery of metal ions from contaminated aqueous medium.

**Supplementary Materials:** The following are available online at http://www.mdpi.com/2304-6740/8/6/40/s1, Figure S1. X-ray diffractogram (a) and Transmission electron microscopy (TEM) (b) of $Fe_3O_4$ nanoparticles. Figure S2. Cadmium weight percentage obtained by EDX analysis of the surface and internal structure of the beads on the adsorption and desorption cycles. Figure S3. Adsorption capacity q (mol/g) at the mono-system (a) and ternary-system (b).

**Author Contributions:** Conceptualization, L.d.C.A. and S.Y.-V.; Methodology, L.d.C.A.; Investigation, L.d.C.A. and S.Y.-V; Writing—original draft preparation, L.d.C.A.; Writing—review and editing, S.Y.-V., Y.P.-R., and J.R.; Supervision, Y.P.-R.; Project administration, J.R.; Funding acquisition, Y.P.-R. and J.R. All authors have read and agreed to the published version of the manuscript.

**Funding:** This research was supported by EP-INTERREG V A (POCTEP) Funds (project NANOEATERS/1378) and by the Consellería de Educación Program for Reference Research Groups project (GPC2017/015 and the Development of Strategic Grouping in Materials—AEMAT at the University of Santiago de Compostela under Grant No. ED431E2018/08, of the Xunta de Galicia.

**Acknowledgments:** The author acknowledges the technical support of the staff at the Ceramic Institute and the research group, from the Department of Analytical Chemistry, Nutrition and Bromatology (Faculty of Chemistry) from the University of Santiago de Compostela.

**Conflicts of Interest:** The authors declare no conflict of interest.

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
