# Peer review of "Efficient Separation of Heavy Metals by Magnetic Nanostructured Beads"

_inorganics, doi:10.3390/inorganics8060040_

Round 1
Reviewer 1 Report
In the manuscript "Efficient separation of heavy metals by magnetic nanostructured beads", the authors present results of investigation of heavy metal removal from polluted waters by adsorption on MAAC. The paper presents an important area of research and the findings are promising; however, they are some key points that should be addressed before the manuscript is accepted for publication:
1) The manuscript is poorly written. The flow and logical development are absent. Actually, the structure of this manuscript is upside down, which makes impossible to read this study! The manuscript must be rewritten with the following structure:
- Introduction.
- Materials and methods
2.1. Synthesis of MAAC beads
2.2. Effect of pH.
2.3. Adsorption studies
2.4. Desorption and reusability
2.5. Characterization of MAAC beads
- Results and discussion
3.1. Characterization of MAAC beads
3.2. Adsorption studies
3.2.1. Effect of pH
3.2.2. Adsorption of mono and tertiary systems
3.2.3. Competitive adsorption evaluation
3.2.4. Adsorption isotherms
3.2.5. Desorption and reusability
- Conclusion
2) This study contains many close similarities to their previous work (Castro Alves et al., 2019, Novel Magnetic Nanostructured Beads for Cadmium(II) Removal). From reviewer's perspective, the present manuscript seems a complement of the findings published by same authors in their previous paper; therefore, please explain the novelty of your study very clear in the Introduction section!
3) In this study, authors re-establish some observations previously made in their previous work (Castro Alves et al., 2019, Novel Magnetic Nanostructured Beads for Cadmium(II) Removal). Only new experimental data must be included in this study! For example, characterization of MAAC was presented also in Castro Alves et al., 2019. Therefore all data regarding characterization of MAAC (SEM, TEM; XRD, FTIR, magnetic properties etc.) that was presented in Castro Alves et al., 2019 must not be repeated in the present manuscript. Furthermore, the following figures should not be included in this manuscript:
- Fig.1C (this manuscript), which is the same with Fig.2A (Castro Alves et al., 2019)
- Fig.1D (this manuscript), which is the same with Fig.2D (Castro Alves et al., 2019)
- Fig.3A (this manuscript), which is the same with Fig.3A (Castro Alves et al., 2019)
- Fig.3B (this manuscript), which is the same with Fig.3B (Castro Alves et al., 2019)
4) Lines 24-27: ” Water pollution by heavy metals has become a serious problem due to the adverse effects on ecosystems and human health. More specifically, Cd, Hg, Pb, or Ni are known to be highly carcinogenic and mutagenic at low concentrations, and may produce acute toxicity or even dead in living organisms, when present slightly above their allowed limits.”
Some references must be added here!
5) Line 155: ” The reusability of the MAAC bead was study by repetitive adsorption and desorption cycles of Cd”
Why was the reusability of the MAAC bead studied only for Cd? Why not also for Hg and Ni?
6) Lines 194-196: ”The higher concentration of hydrolyzed ions like H+ and H3O+ present in solution at pH 4.5, may allow an enhanced binding of Hg (II) metal ions to the sodium alginate surface”
First: H+ and H3O+ is one and the same ion; H3O+ is the hydrated form of H+;
Second: At pH 2 concentration of H+ is much higher than at pH 4.5; then, according to your explanation, binding of Hg should take place much better at pH 2, but it doesn’t. You should try giving a better explanation for the high retention of Hg at pH 4.5!
Author Response
Please see the attachement

Reviewer 2 Report
In the presented manuscript: “Efficient separation of Heavy metals by magnetic nanostructured beads” by Lisandra de Castro Alves et al., the authors describe the application of magnetic beads. The subject of submitted paper is in general important but the presentation does not sufficiently describe therefore some improvements are needed.
- There is a lack of information about why the presented study concerns nanostructured beads.
- The presented TEM and SEM images are not sufficient to convince the reader about nanostructures' presence.
- There is some mess in Fig. 3. Not needed numbers appear very randomly.
- There is an estimation of the iron oxide content in the whole composite?
- What is the size of nanoparticles?
- There is any dependence on adsorption efficacy caused by Fe-oxide content or particle size?
Reviewer 3 Report
The paper reports about the characterization and adsorptive properties of alginate beads containing activated carbon and manetic nanoparticles. The work has several drawbacks, problems in the experimental design and interpretation of the results. Hence, for the reasons described below the paper cannot be accepted for publication.
Page 2, TEM. TEM should provide images with a much larger magnification and resolution than SEM images. Actually, clusters of magnetite nanoparticles should be visible. Here there is no reason to show the TEM image at such low magnification.
Page 3, Raman spectrum. It does not add any useful information. A comparison of the spectrum of the beads with the spectra of activated carbon and magnetite should be shown.
Page 3. Line 108. The band attributed (correctly) to Fe3O4 is not due to “stretching vibrations of Fe-O bond in magnetite” since there are not Fe-O bonds! The band is due collective vibrations of the magnetite lattice.
Page 3. XRD. The authors should estimate the size of the crystallite of magnetite nanoparticles by analyzing the width of the diffractions peaks
Page 4, Magnetic properties. Since the magnetization is per g of Fe3O4, how was the content of magnetite in the beads determined?
Page 5, eq. 1 and line 161. Something is wrong in this definition. If q is mg/g then M should be a concentration (mg/L) and V in L
Page 5, Effect of pH. The formation of hydroxo complexes (and possibily the precipitation of hydroxides) on the adsorption capacity for the three metal ions as a function of pH should be considered in order to interpret the observed trends.
Page 6, line 228. It is obvious that the removal efficiency decrease upon increasing the initial concentration since the quantity of adsorbed metal ions is reaching a saturation.
Page 7. In the caption of figure 7a and b the pH at which the measurements were carried out should be reported.
Page 7, table 2. According to the definition of the “evaluation ratio” (5) it is not clear what is the meaning of the last raw in table 2. The values of E for all metals do show a random behaviour as a function of the initial concentration. The only conclusion which can be drawn from these data is that for Hg, E is less than 1 at all concentrations.
Page 7. Problem with concentration. The authors express the concentration as mg/L. It would be better to use molarity. In particular when discussing the preferential adsorption in the mixed solution it is quite different to base the interpretation of the results on mg/L or molarity since the atomic weight of Hg is almost two times that of Cd and four times that of Ni. Hence, it makes more sense to use molar concentrations rather the weight concentrations.
Page 8. Adsorption isotherms. Line 287. What is the meaning of “unit equilibrium concentration”?
Line 288: “1/n indicates the adsorption intensity of metal ions”. What does it mean?
Page Table 3. On the basis of the R2 values it is not possible to conclude that the Freundlich isotherm describes better the experimental data than the Langmuir isotherm for the single metal ion solutions. For the mixed solution, the linear correlation for the Freundlich isotherm seems to be better than for the Langmuir isotherm but I wonder how it was possible to fit experimental data like those in A2 which exhibit maxima with those isotherms.
Page 9, line 337. “under magnetic agitation”. A curiosity: by using magnetic stirring did not the alginate beads containing magnetite nanoparticles stick to the magnetic rod?
Reviewer 4 Report
Review
Manuscript No: inorganics-784625
GENERAL COMMENTS:
The manuscript reports an interesting application of novel material synthetized from magnetite, alginate and activated carbon. The adsorption ability of material for separation of selected heavy metals in a form of single-elemental and multi-elemental solutions is presented.
Generally, the idea and the whole concept of work are rather acceptable. The majority of obtained results are well described and discussed. The best part of the manuscript denotes to characterization of novel material.
Although, there are some weaknesses, especially in adsorption study performance and conclusions.
The first problem is high concentration range of examined metals used in this study (10-250 mg / L). If your concern is related to water and wastewater management (please consult EU Water Framework Directive documents!), you’ll find that such range is over than a magnitude higher when compared to usual metals occurrence in such systems! Therefore, the requisite of 14 mg of adsorbent (not explained in the text!) in such concentrated solutions could not be related to water cleaning procedures.
Another major obstacle here is the behaviour of Hg (II) ion in solutions and its measurement by ICP-AES technique. It is already known that mercury concentrations higher than few micrograms per millilitre caused significantly pronounced memory effects during ICP measurements. The mercury release from glass parts of plasma instrument cannot be controlled. Therefore, the signal is not uniformed throughout the measurements. Such effects are additionally more complex if the solution matrix contains chloride ions, even in trace quantities (as was in your experiments). Precipitation of mercury-chloride affect the measurable mercury content. For this reason, mercury determination by ICP is performed at very low conc. levels (ppb). The most appropriate technique for Hg determination would be mercury-analyser. Your conclusions of unpredictable Hg behaviour are correct, but it was based on poorly settled experimental conditions! My suggestion is to eject this part that denote to mercury sorption, because it is failed due to erroneous measurement.
The kinetic experiments are well established. However, the explanation of differences in electrochemical potential of Cd and Ni at magnetite surface might be helpful for understanding the observed effects.
Specific comments:
Title:
„Heavy“ into „heavy“ ; lowercase letter!
Introduction:
Page 1 Line 39 chemical versatility
Results and Discussion:
Page 4 Line 140-141 assynthesized?
Page 5 Line 192 ...at higher pH values.
Page 6 Line 207 3.3. Adsorption of mono-metal and ternary (or three-metal)...
Line 209 ternary system
Line 219 V (italic!) is the volume ... M (italic!) is the weight
Line 223 the ternary...
Line 224 same as above
Line 225 nearly twice (Higher? Lower?)
Line 226 ...it can be observed uniform (or balanced) increase...
Line 231 ..in the adsorption experiment with ternary system,...
Page 7 Line 246 ternary (and throughout the text!)
Line 251 ...in the mono-metal system
Line 251-255 E (italic!)
Page 8 Line 271 200 mg/L or mg L-1
Line 286 The value KF can... (Not begin the sentence with abbreviation or symbol!)
Line 288 same as above
Materials and Methods:
Page 9 Line 319 previous study
Line 336 pH values ( Did you adjust the values by use of buffer solution ?)
Line 342 Explanation for exploiting of 14 mg of adsorbent is missing!
Conclusions:
Page 10 Line 390-391 ...„,although more exhausted studies are needed to understand the adsorption trend of Hg (II) metal ions“... (to delete this statement!)
References:
To check journal abbreviations! For example, Ref [23]!
Author Response
Please the attachment

Round 2
Reviewer 1 Report
You were right that the manuscript sections are disposed correctly, according to the Inorganics format. Please excuse me! However, it is a strange format!
All the best andt stay safe and healthy!
Author Response
The authors want to acknowledge all the corrections suggested by the Referees , that have allowed us to improve the text.
“You were right that the manuscript sections are disposed correctly, according to the Inorganics format. Please excuse me! However, it is a strange format! All the best and stay safe and healthy!”
We agree with the Referee, the format of the journal seems strange. The authors thank the suggestions and questions raised by the reviewer, which have served us to improve the manuscript. All the best.

Reviewer 2 Report
In the presented manuscript: “Efficient separation of Heavy metals by magnetic nanostructured beads” by Lisandra de Castro Alves et al., the authors describe the application of magnetic beads. The subject of submitted paper is in general important but the presentation does not sufficiently describe therefore some improvements are needed.
- Still there is no proof that authors have any nanoparticles in the system at all. Very miraculous was changed scale from micro to nanometers between versions of the paper.
- Why authors did not estimate any error bars for the presented data?
- It is a real jump of the values for Hg (Fig.6)?
- There is an estimation of the iron oxide content in the whole composite?
- Where is the proper TEM image of the nanoparticles?
- There is any dependence on adsorption efficacy caused by Fe-oxide content or particle size?
7. .
Author Response
Dear Reviewer,
Please find attached the answer to your questions. Hope it clarifies you. All the best.
---------------
The authors thank the suggestions and question raised by the reviewers, which have served us to improve the manuscript.
Following his/her comments we have made the following improvements and changes:
- “Still there is no proof that authors have any nanoparticles in the system at all. Very miraculous was changed scale from micro to nanometers between versions of the paper.”
Along the manuscript, different results were presented confirming that magnetite nanoparticles used in the synthesis procedure were effectively embedded in the MAAC beads:
- the X-Ray diffraction pattern presented in the manuscript corresponds to the analysis of MAAC beads, and the sharp diffraction peaks appearing in the diffractogram, correspond to magnetite NPs,
- -the magnetization cycle , presented in figure 4b, showing the superparamagnetic behaviour of magnetite NPs ( prior to be used in the synthesis) together with the magnetization of the composite MAAC beads, which is also superparamagnetic, confirming that magnetite NPs stand inside the polymer matrix without suffering aggregation.
- TEM image of magnetite NPs used to prepare the magnetic beads was already included in the supporting file, Annex, in figure A1., together with the diffractogram of pure magnetite NPs, used in the synthesis.
Regarding the change of scale and images in Figure 1, this was required in the first round of revisions, by Referee 3, and accordingly we modified some of the SEM and TEM images, to include others with larger magnification in the revised article.
- “Why authors did not estimate any error bars for the presented data?”
Following his/her suggestion it was included the standard deviation on the subtitles of the Figures 5 and 6.
- “It is a real jump of the values for Hg (Fig.6)?”
We ascribe the jumps in figure 6, to mercury speciation, and the protonated and unprotonated species of alginate, that are created at different pH. However, to clarify this point in deep, we will conduct further studies, involving also other specific aspects regarding the role of Hg in the ternary experiments, that seems to modulate the trend of adsorption of Cd and Ni. Unfortunately, due to the present pandemic closure, we cannot carry on any experiment currently, and these tasks will be done when our facilities will be reopened, and will be reported in a future work.
- “There is an estimation of the iron oxide content in the whole composite?”
The content of iron oxide mass in the MAAC beads was determined to be 23% using thermogravimetry analysis, and following his/her suggestion this clarification is included in the section 2.4 , lines 136-137.
- “Where is the proper TEM image of the nanoparticles?”
On Appendix (Figure 1A) it was included a TEM image of magnetite nanoparticles.
- “There is any dependence on adsorption efficacy caused by Fe-oxide content or particle size?”
The main interest of including superparamagnetic magnetite NPs is to obtain a lightweight material with a sharp magnetic response under the application of external magnetic fields, capable of being magnetically stirred and extracted from the liquid matrix. In this regard, the use of superparamagnetic magnetite NPs with no remanence, is crucial to avoid magnetic aggregation problem between the beads, which arises when using large magnetite particles with remnant magnetization.
Moreover, different beads were prepared using varying contents of pure magnetite NPs, with no functional coating shell, and their adsorption capacity was tested showing no substantial influence on the adsorption efficacy.

Reviewer 3 Report
The paper can be accepted after amendment considering the following points
- “Page 3, Raman spectrum. A comparison of the spectrum of the beads with the spectra of activated carbon and magnetite should be shown.”
We have only the Raman measurements for the commercial activated carbon (Figure 2, page 3), but not for the beads or magnetite. Currently due to the pandemic state it is impossible for us to do the rest of the measurements, but we have this consideration for further works. However, we have completed the FTIR spectra (Figure 3b, page 3) with the magnetite spectra.
As it is now, the spectrum is useless. So I suggest to remove it
- “Page 7, table 2. According to the definition of the “evaluation ratio” (5) it is not clear what is the meaning of the last raw in table 2.”
The last raw in table 2 is the sum evaluation ratio of all metals in the tertiary system, whereas the previous rows are related to the individual evaluation ratio of each metal in the tertiary system. (page 8, table 2).
What do the sums of the evaluation ratios for the three ions in the mixed solutions tell us? The author should discuss this point
- “Page 7. Problem with concentration. The authors express the concentration as mg/L. It would be better to use molarity.”
To study the adsorption mechanism is more convenient to use the concentration as mg/L, as the amount of metal ions adsorbent per mass unit of adsorbent at equilibrium is given (qe mg/g).
The authors fail to answer to my comment since they did not consider the last part of my comment:
In particular when discussing the preferential adsorption in the mixed solution it is quite different to base the interpretation of the results on mg/L or molarity since the atomic weight of Hg is almost two times that of Cd and four times that of Ni. Hence, it makes more sense to use molar concentrations rather the weight concentrations.
Author Response
Dear Reviewer,
Thank you for your suggestions, please find attached the answer to your questions.

Reviewer 4 Report
Dear Authors,
Thank you for your comments on general remarks. I am generally satisfied with your answers.
Besides general remarks you could find the set of Specific comments where I inserted some specific tasks that should be improved throughout the text. I didn't find that you inserted corrections according to my Specific claims!
For example, word Heavy in the title should be written as heavy; tertiary should be changed into ternary or three-component, etc.
Therefore, I suggest you to check once again the list of Specific comments that was already sent.
Author Response
Dear Reviewer,
Thank you for your corrections and valued comments, we say sorry to not correct your previous remarks. Please find attached the answer to your questions.

Round 3
Reviewer 3 Report
Accept